# Humoral Immunity across the SARS-CoV-2 Spike after Sputnik V (Gam-COVID-Vac) Vaccination

**DOI:** 10.3390/antib13020041

**Published:** 2024-05-11

**Authors:** Alejandro Cornejo, Christopher Franco, Mariajose Rodriguez-Nuñez, Alexis García, Inirida Belisario, Soriuska Mayora, Domingo José Garzaro, José Luis Zambrano, Rossana Celeste Jaspe, Mariana Hidalgo, Nereida Parra-Giménez, Franklin Ennodio Claro, Ferdinando Liprandi, Jacobus Henri de Waard, Héctor Rafael Rangel, Flor Helene Pujol

**Affiliations:** 1Laboratorio de Bioquímica Celular, Centro de Microbiología y Biología Celular, Instituto Venezolano de Investigaciones Científicas (IVIC), Caracas 1020A, Venezuela; cornejo.alejandro@gmail.com; 2Laboratorio de Virología Celular, Centro de Microbiología y Biología Celular, IVIC, Caracas 1020A, Venezuela; chrfranco.94@gmail.com (C.F.); jlzr.lab@gmail.com (J.L.Z.); 3Laboratorio de Virología Molecular, Centro de Microbiología y Biología Celular, IVIC, Caracas 1020A, Venezuela; rodriguez95mariajose@gmail.com (M.R.-N.); dgarzaro@gmail.com (D.J.G.); rossanajaspesec@gmail.com (R.C.J.); hrangel2006@gmail.com (H.R.R.); 4Instituto de Inmunología, Universidad Central de Venezuela (UCV), Caracas 1040A, Venezuela; alexisgarcia27@gmail.com (A.G.); ibelisariogomez@gmail.com (I.B.); sori_mayo@hotmail.com (S.M.); 5Laboratorio de Inmunoparasitología, Centro de Microbiología y Biología Celular, IVIC, Caracas 1020A, Venezuela; mariana.hidalgo.r@gmail.com; 6Laboratorio de Fisiología de Parásitos, Centro Biofísica y Bioquímica, IVIC, Caracas 1020A, Venezuela; bionereida@gmail.com; 7Departamento de Tuberculosis, Servicio Autónomo Instituto de Biomedicina “Dr. Jacinto Convit”, UCV, Caracas 1010A, Venezuela; frank241293@gmail.com (F.E.C.); jacobusdeward@gmail.com (J.H.d.W.); 8Laboratorio de Biología de Virus, Centro de Microbiología y Biología Celular, IVIC, Caracas 1020A, Venezuela; fliprand@gmail.com; 9Laboratorios de Investigación, Facultad de Ciencias de Salud, Universidad de Las Américas (UDLA), Quito 170125, Ecuador

**Keywords:** COVID-19, SARS-CoV-2, antibody, ELISA, spike, S1, S2, RBD, epitopes, neutralization

## Abstract

SARS-CoV-2 vaccines have contributed to attenuating the burden of the COVID-19 pandemic by promoting the development of effective immune responses, thus reducing the spread and severity of the pandemic. A clinical trial with the Sputnik-V vaccine was conducted in Venezuela from December 2020 to July 2021. The aim of this study was to explore the antibody reactivity of vaccinated individuals towards different regions of the spike protein (S). Neutralizing antibody (NAb) activity was assessed using a commercial surrogate assay, detecting NAbs against the receptor-binding domain (RBD), and a plaque reduction neutralization test. NAb levels were correlated with the reactivity of the antibodies to the spike regions over time. The presence of Abs against nucleoprotein was also determined to rule out the effect of exposure to the virus during the clinical trial in the serological response. A high serological reactivity was observed to S and specifically to S1 and the RBD. S2, although recognized with lower intensity by vaccinated individuals, was the subunit exhibiting the highest cross-reactivity in prepandemic sera. This study is in agreement with the high efficacy reported for the Sputnik V vaccine and shows that this vaccine is able to induce an immunity lasting for at least 180 days. The dissection of the Ab reactivity to different regions of S allowed us to identify the relevance of epitopes outside the RBD that are able to induce NAbs. This research may contribute to the understanding of vaccine immunity against SARS-CoV-2, which could contribute to the design of future vaccine strategies.

## 1. Introduction

Four years have passed since the World Health Organization (WHO) declared the COVID-19 pandemic due to SARS-CoV-2 on 11 March 2020. More than 770 million cases of COVID-19 and over 7 million deaths have been officially reported since then, although it is estimated that these numbers are considerably higher [1]. Nonetheless, SARS-CoV-2 vaccines have contributed to attenuate such a burden by promoting the development of effective immune responses, thus reducing the spread of the pandemic, the severity of the disease, hospitalizations and deaths [2].

SARS-CoV-2, which belongs to the family *Coronaviridae*, is an enveloped virus with a positive sense genome of around 30,000 nt. The genome codes for four structural proteins (nucleocapsid or N, spike or S, membrane or M and envelope or E), 15 nonstructural proteins and 8 accessory proteins [3]. The structural homotrimeric glycoprotein S has been used as the target for many of the vaccines developed [4,5]. This protein is composed of S1 and S2 subunits. The surface subunit S1 is composed of 672 amino acids and is organized into four domains: an N-terminal domain (NTD), a C-terminal domain (CTD, also known as the receptor-binding domain, RBD) and two subdomains (SD1 and SD2) [5]. The highly antigenic region known as the receptor-binding domain (RBD) mediates the interaction with the receptor angiotensin converting enzyme (ACE2) and the binding of the majority of neutralizing antibodies [6]. It is known that other epitopes outside the RBD, like the NTD, are also important in immunity and contribute to the antigenic profile of the S protein [7,8,9], but the effects of antibody recognition are not yet well characterized. The transmembrane subunit S2 is composed of 588 amino acids and contains a hydrophobic N-terminal fusion peptide (FP), two heptad repeats (HR1 and HR2), a transmembrane domain (TM) and a cytoplasmic tail (CT), the S2 subunit being the more conserved among all coronaviruses [10,11]. The S antigen was used as the immunogen in several vaccine constructs.

Vaccine strategies included nonreplicating adenoviral vectors, nucleic acid (mRNA), whole inactivated viruses and protein subunit-based vaccines. The two-component adenovirus vector vaccine Gam-COVID-Vac (Sputnik-V) was the second-most distributed vaccine in Venezuela and was also employed in other Latin American countries [12,13]. Despite being distributed in at least 35 countries and having over 1.3 billion doses administered by March 2024 (according to ourworldindata.com, accessed on 13 March 2024), few studies are available on Sputnik-V efficacy and immunogenicity compared to other vaccines. In order to deepen the understanding of the effectiveness and immunogenicity of Sputnik-V, the aim of this study was to explore the antibody reactivity of individuals vaccinated with Sputnik-V towards different regions of S. Neutralizing antibody (NAb) activity was assessed and correlated with the reactivity of the antibodies to S over time.

## 2. Materials and Methods

### 2.1. Study Design, Participants and Serum Recollection

The samples evaluated were obtained from volunteer subjects belonging to Gam-COVID-Vac (NCT04642339), a prospective, double-blind, randomized, placebo-controlled clinical trial in Venezuela, from December 2020 to July 2021. Inclusion criteria included male and female volunteers, aged 18 years or older, who read and signed the informed consent form and tested negative for HIV, hepatitis and syphilis. Volunteers had no clinical history of COVID-19 and tested negative for both IgG and IgM antibodies to SARS-CoV-2 by enzyme immunoassay (PISHTAZTEB Diagnostic, Tehran, Iran) and negative for SARS-CoV-2 by RT-PCR. Exclusion criteria were receiving any vaccination/immunization within 30 days prior to enrollment, use of steroids (except hormonal contraceptives) and/or immunoglobulins or other blood products within 30 days before enrollment, completion of immunosuppressive therapy within 3 months prior to enrollment, pregnancy and lactation. The inclusion and exclusion criteria details are shown in more detail on the clinical trials website (clinicaltrials.gov/ct2/show/NCT04642339 accessed on 13 March 2024). The volunteers were from Caracas, ranging in age from 20 to 80 years old. A total of 133 sera were obtained from 80 male and 53 female volunteers, divided between vaccinated (*n* = 105) and placebo controls (*n* = 28). All volunteers received a first dose on day 1 and a second dose 21 days after the first dose. Samples were taken 42 days post vaccination (dpv), counting after the first dose and 21 days after the second dose. For a subgroup of 46 vaccinated volunteers, sera were also obtained at 180 dpv. The trial and all its procedures were approved by the National Ethics Committee for Research in COVID-19.

### 2.2. Antigens

The recombinant antigens for the enzyme-linked immunosorbent assays (ELISA) were acquired from MyBioSource Inc. (San Diego, CA, USA). The ancestral S protein (MBS8574721), S1 (MBS8309640), S2 (MBS9141947), RBD (MBS8574741), and N (MBS5316649) antigens were used. The concentration of antigens was determined using a Qubit™ Protein Assay according to the manufacturer’s specifications (catalog number Q32866, Thermo Fisher, Waltham, MA, USA).

### 2.3. ELISA Reactivity

To determine the reactivity of the sera towards the different regions of S and N, the latter to rule out a SARS-CoV-2 infection, the microtiter plates were sensitized with 1 µg/mL (2 µg/mL for N) of the antigens in a final volume of 50 µL, and incubated at 4 °C overnight. The next day, the solution was decanted and the plates were incubated with 150 µL of 1× blocking solution (Abcam, Cambridge, UK, ab126587) for at least 1 h at room temperature. During this incubation, the samples of vaccinated and controls were diluted 1/100 in 1× blocking solution. After the blocking time, the plate was washed 6 times with 0.01% PBS-Tween, 100 µL of the diluted serum samples was incubated for 2 h at 37 °C and the plate was washed again. The antihuman IgG secondary antibody conjugated to horseradish peroxidase (Jackson ImmunoResearch Inc., West Grove, PA, USA) diluted 1/70,000 was added and incubated for 1 h. A chromogenic substrate solution of peroxidase, TMB (3,3′,5,5′-tetramethylbenzidine), was used for the color development reaction. To stop the reaction, 50 µL of HCl [3M] was used. For the reading and recording of the data, a spectrophotometer (SpectraMax 250, Hampton, NA, USA) was used at 450 nm. As negative controls, 18 sera from apparently healthy individuals obtained before the pandemic were used. A positive control to normalize data was used, consisting of a mixture of two sera from highly responsive individuals vaccinated with Sputnik-V and with two symptomatic infections confirmed by RT-PCR. An additional 20 sera with hybrid immunity (symptomatic infection(s) plus Sputnik-V vaccination), with infections confirmed by RT-PCR, were included. Previous reports show that hybrid immunity results, on average, in higher antibody titer and higher neutralizing activity compared to fully vaccinated individuals without prior COVID-19 [14,15]. Reactivity against SARS-CoV-2 antigens (ancestral: S, S1, S2 and RBD) was assessed within this group of 134 individuals. The optical densities (O.D.s) of the blank were subtracted from the O.D.s of the samples. O.D.s exceeding the cutoff established with negative control mean plus 3 standard deviations were considered as responders to N or S. Relative levels of IgG antibodies were normalized as the sample-to-positive ratio (S/P) [12,16] using the following formula, with the respective controls assayed on each ELISA plate: S/P = ((O.D. sample − O.D. negative controls)/      (O.D. positive controls − O.D. negative controls)) × 100.

The BAU (binding antibody units) serum international standard from the World Health Organization (WHO) was used for some assays to establish a standard curve and measure the relative concentration of antibodies in each sample. 

### 2.4. Surrogate Neutralization Test (SNT) Based on ACE2 Blocking Adsorption Immunoassay 

A commercial kit that allows the detection of neutralizing antibodies through a competitive ELISA (SARS-CoV-2 Neutralization Antibody Detection Kit, Elabscience^®^, Houston, TX, USA) was used according to its manufacturer’s instructions. Briefly, the test quantifies NAbs against the RBD that are able to block the interaction between S and ACE-2. In this assay, 50 µL of the sera diluted 1/10 was incubated with 50 µL of the solution containing the recombinant spike protein (RBD) conjugated to HRP in a microtiter plate already sensitized with ACE2 for 1 h. Subsequently, the plate was washed 3 times with the commercial washing solution provided by the kit and 90 µL of commercial substrate from the commercial kit was added for a 15-min incubation. After this time, 50 µL of the STOP solution provided by the kit was added. For the reading and recording of the data, a spectrophotometer (SPECTRAmax 250) was used at 450 nm. If neutralizing antibodies are present in the sample, they will inhibit spike interaction with recombinant ACE-2, which will result in a decrease in the optical density signal. The commercial kit provides a titration curve for assessing the titers of NAbs anti-RBD.

### 2.5. Plaque Reduction Neutralization Test

A plaque reduction neutralization test (PRNT) was conducted on 20 samples of serum collected at 42 days post-vaccination (dpv) and 18 samples from 180 dpv. PRNT was performed according to a previously reported procedure [17]. VERO C1008 cells (Vero 76, clone E6, vero. ATCC, Manassas, VA, USA), were maintained at 37 °C and 5% CO_2_ in RPMI medium (Thermo Fisher Scientific, 11875093) supplemented with 10% fetal bovine serum (FBS, Thermo Fisher Scientific, 16000044) and 1% penicillin–streptomycin (Thermo Fisher Scientific, Waltham, MA, USA). Infection was performed in a biosecurity cabinet within a level 3 biosecurity facility. Viral seeds of a SARS-CoV-2 ancestral strain (B.1.1.3) were incubated for 30 min at 37 °C with different dilutions of vaccinated and control sera. The infection process was performed for one hour at 37 °C in a 5% CO_2_ atmosphere, and the cells were washed twice with PBS to remove the noninternalized viruses. The cells were overlayed with 0.5% carboxymethyl cellulose in a culture medium and incubated for 72 h. Then, the cells were fixed with 4% formaldehyde and stained with crystal violet to count the number of lytic plaques under each condition. The PRNT50 of each serum was determined through a nonlinear regression test, defining this value as the reverse of the dilution at which 50% of the virus is neutralized. 

### 2.6. Statistical Analysis

All statistical tests and graphs were performed with PRISM GraphPad© (San Diego, CA, USA). For the analysis of the reactivity toward the ancestral spike protein and its different regions, the Kruskal–Wallis test with an alpha value of 0.05 was used, with its respective Dunn’s correction post hoc analysis. For the analysis of reactivity toward the spike and its different regions and SNT comparisons, a Mann–Whitney test with an alpha value of 0.05 was used. For ELISA and PRNT50 comparisons at 42 and 180 dpv, a paired Wilcoxon test was used, with an alpha value of 0.05. To determine the degree of association between IgG antibody reactivity, SNT and PRNT50, the correlations matrix test was performed using the Spearman correlation coefficient with a confidence interval (CI) of 95%. For frequency comparison among groups, statistical difference was assessed with the Chi-square test (Epi InfoTM, Centers for Disease Control and Prevention, Atlanta, GA, USA).

## 3. Results

### 3.1. Seroconversion Rates in Non-Exposed Vaccinated Individuals in the Clinical Trial

A total of 133 volunteer sera were obtained from the Gam-COVID-Vac clinical trial in Venezuela. During the development of the clinical trial, no serious adverse events were reported. All adverse events reported were mild and resolved spontaneously without sequelae. Of these, 105 were from vaccinated individuals and 20/106 exhibited reactivity against N, suggestive of exposure (Figure 1). The sera of 85/105 vaccinated (with no serological evidence of exposure) were analyzed for their reactivity against different regions of S. Figure 2 shows the levels of Abs (BAU/mL) for S in the sera from the different groups. A total of 92% of the vaccinated non-exposed individuals could be classified as responders according to their BAU titer. The mean BAU/mL value of the sera from the vaccinated volunteers was significantly higher than the mean BAU/mL value of the placebo group. However, some sera (52%) from placebo individuals without reactivity to N showed reactivity to S in this assay (Figure 2), suggesting that some of them were also exposed during the course of the clinical trial or before it. It is important to note, however, that some of the sera apparently reactive to N or S exhibited very low S/P or BAU values near the limit of positivity, suggesting some unspecific reactivity.

### 3.2. Reactivity to Different Regions of SARS-CoV-2 S 

In order to dissect the humoral immune response to S, reactivity to the different regions of this protein was evaluated (Figure 3). The reactivities of sera from apparently healthy individuals collected before the pandemic (prepandemic) were also analyzed as negative controls. S and S1 regions were recognized at a similar frequency, while fairly less sera recognized the RBD region and significantly less sera recognized the S2 domain (58%, *p* < 0.001) (Figure 3a). For vaccinated individuals at 42 dpv, statistically significant differences were observed (*p* < 0.0001) in the S/P average ratio between the different antigens. Multiple comparisons revealed that the S1 region was recognized with a greater S/P value in the sera than other regions (*p* = 0.0054), while S and the RBD exhibited similar reactivity and S2 was recognized with significantly lower reactivity (*p* < 0.0001).

The reactivities of the vaccinated volunteers were compared with those with serological evidence of exposure, suggestive of hybrid immunity (Figure 3b). The average reactivity of the sera from the group of individuals with hybrid immunity was significantly higher than that of the vaccinated-only group (Wilcoxon test), especially in the case of the S2 region, where a more pronounced difference was observed (4.1×). Reactivity of some sera with S (*n* = 2), S2 (*n* = 8) and the RBD (*n* = 2) was observed in the 20 prepandemic sera, the S2 region being the most frequently recognized (*p* < 0.0076) (Figure 3c). These samples were excluded from the antigen cutoff calculation. The sera at 180 dpv (in vaccinated individuals who completed the clinical trial, *n* = 21) exhibited a significant decrease in reactivity to S and the RBD (Wilcoxon paired test, *p* < 0.05), while no significant differences were observed for S1 and S2. In some cases, an increase in S/P value was observed at 180 dpv (Figure 3d).

### 3.3. Surrogate Neutralization Test (SNT) 

The SNT based on the ACE2 blocking immunoassay allowed us to quantify the NAbs against SARS-CoV-2 directed against the RBD (Figure 4). Significant differences were observed between the placebos and the vaccinated, as expected. No significant differences were observed between the NAb levels at 42 and 180 dpv. The prepandemic sera did not show NAbs by this test, even those that exhibited ELISA reactivity to some S regions.

The ELISA reactivity to different regions of S was correlated with the demographic parameters (Figure 5). A low negative but significant correlation was found between reactivity to S and the level of SNT with age. This was not observed for S1, S2 and the RBD (not significant). No significant correlation was found between reactivity to S and its regions and sex n. Significant positive correlations were found between reactivity to S and to S1 and the RBD but not to reactivity for S2. The highest positive correlation was found between SNT and reactivity to the RBD, as expected, with the lowest correlation between SNT and reactivity to S2.

### 3.4. Plaque Reduction Neutralization Test (PRNT50)

Figure 6 shows the PRNT50 for the sera analyzed at 42 and 180 dpv. No PRNT50 titers were observed for the placebo sera. No significant difference was observed between the PRNT50 values at 42 and 180 dpv. Some individuals even exhibited higher neutralization titers at 180 dpv. The PRNT50 titers were correlated with the other parameters analyzed in this study (Figure 7). The highest correlation was obtained with reactivity to S1, followed by S and the RBD. A lower correlation was observed between SNT and reactivity to S2. All comparisons showed moderate to low correlations with PRNT50. Similar results were observed when analyzing sera at 180 dpv (Figure 8), with a reduction in the correlation with reactivity to the RBD.

## 4. Discussion

The dissection of Ab reactivity to different S regions allowed us to correlate this reactivity with the neutralizing ability of sera from volunteers vaccinated with Sputnik V. Determining exposure to SARS-CoV-2 by analyzing reactivity to N was essential to exclude from the study those individuals who were exposed to the virus during the course of the clinical trial. The first observation is that a quite important proportion of volunteers exhibited evidence of SARS-CoV-2 exposure (anti-N Abs) during the clinical trial. It should be noted that anti-N Abs have a low persistence in serum compared to anti-S ones [18]. In fact, the analysis of the reactivity to S revealed that additional volunteers from the placebo group exhibited anti-S Abs without reactivity to N, suggesting that an even higher degree of exposure in the group might have occurred. The apparent relatively high incidence of exposure to the virus during the window of the clinical trial could be explained by the increase in the number of cases in Venezuela during the period of the clinical trial [19]. It is important to note that only two of the apparently exposed volunteers, both from the placebo group, reported a symptomatic COVID-19 episode. This suggests that the Ab reactivity to SARS-CoV-2 proteins might have been associated with asymptomatic or mild exposure in most of the cases.

Of the total vaccinated, 92% of individuals responded to vaccination by generating detectable anti-S Abs, a frequency similar to that reported in previous studies, and in agreement with the high efficacy and seroconversion rate reported for the Sputnik vaccine [20,21,22,23]. It has been reported that BAU levels between 13 and 141 BAU/mL provided only 12.4% protection against SARS-CoV-2 (ancestral strain); a concentration between 141 and 1700 BAU/mL 89.3% protection and a concentration of 1700 BAU/mL and higher provided complete protection [24]. In our study, a value less than 119 BAU/mL was considered the threshold for defining nonresponder individuals. 

The analysis of Ab reactivity to the different regions of S showed that S1 was the most important antigenic region, exhibiting higher recognition than the RBD, while S2 turned out to be the region with the lowest reactivity. These results are consistent with previous reports describing other regions within S1 outside the RBD that are capable of generating an antibody response [25]. On the other hand, the fact that S1 exhibited greater antigenicity than S in our assays also suggests that some of these sites are probably encrypted or less exposed when S is in its homotrimeric conformation or prefusion states [25,26,27]. Most of the S in the SARS-CoV-2 virions are present in a metastable prefusion conformation, although some S can adopt an extended postfusion rod-like conformation due to the premature dissociation of S1 from S2 (independent of the interaction with ACE-2), which also releases the soluble S1s subunit. S1s can act as an independent immunogen and probably induces a B lymphocyte response independent of T through cross-linking of the B cell receptor (BCR) [28] in addition to the canonical mechanisms of B lymphocyte stimulation. This same conformational phenomenon has been reported in the S product of the Sputnik-V vaccine construct, which does not have stabilization mutations [29]. The role of the dissociation of the S1 subunit in immunity or the pathophysiology of the disease is still unclear. Some data suggest that it may contribute to a higher proportion of no NAbs compared to neutralizing ones [30]. We hypothesize that it may contribute to the generation of Abs against regions outside the RBD, such as the NTD. It has been described that dissociated S1 subunits are capable of forming immune complexes with Abs [30] that can probably activate Fc receptors in dendritic and follicular cells, promoting the S1 antigen presentation. Additionally, other studies have indicated that the S1 subunit, particularly the NTD, can activate cells within the innate immune system [31].

The S2 region triggers the fusion of the viral membrane with the cellular membrane. Some authors have suggested that it is a less immunogenic region than S1 because of its high degree of glycosylation [32,33]. In contrast to the low antigenicity exhibited by S2 in the sera of vaccinated volunteers, this was the region of S most frequently recognized by prepandemic control sera (Figure 3c). The presence of these antibodies is probably due to the seasonal prevalence of the four common types of human coronaviruses, especially OC43 and HKU1 [34,35]. Some authors have described, through microarray assays with recombinant S peptides, that there are immunogenic sites in S2 between amino acids 818 and 835, the region corresponding to the fusion peptide (FP) that is highly conserved among coronaviruses [26,36,37]. Targeting the FP with antibodies can prevent the host protease TMPRSS2 from cleaving the S2 subdomain, thus reducing viral entry [36]. In addition, the amino acid sequence of FP does not vary among the variants of SARS-CoV-2 currently known [38]. Interestingly, when comparing Ab reactivity to the different S regions between the sera of vaccinated and (probably) exposed vaccinated volunteers, the highest increase in reactivity was found for the S2 region (4.1×). This observation might suggest that natural infection induces a stronger reactivity to S2 than the one elicited by the Sputnik V vaccine. The results obtained coincide with other reports, such as those of Polvere et al., who identified that natural infection by COVID-19 combined with vaccination results, on average, in a higher antibody titer and greater neutralizing activity with respect to fully vaccinated people without a history of COVID-19 [14].

No differences were observed between sex and anti-S Abs levels, in agreement with previous reports [39]. The correlation between reactivities to S, SNT and age exhibited an age-independent response. This has also been reported in other Sputnik-V studies [20,22,40]. While a reduction in the total Abs was observed by ELISA, the levels of NAbs, measured by SNT or PRNT50, were maintained on average at 180 dpv, suggesting a maturation process in the immune response. In contrast, a decline in NAbs has been reported for mRNA vaccines [41]. Some low-level transgene long-term expression has been described [42,43]. The persistence of antigen expression may be a distinctive feature of adenovirus vector vaccines. It has been proposed as contributing to the induction of sustained immune responses and lasting immunity to S and could explain the persistence of NAbs. Another mechanism, such as the maturation of B lymphocytes in germinal centers (GCs) in the lymph node [44,45] has also been reported as contributing to the diversification of response and maturation of antibody affinity [46,47,48].

SNT levels were not significantly correlated with PRNT50 titers. Moreover, at 42 dpv, the highest correlation with PRNT50 was observed with S1 reactivity. This observation is in agreement with the evidence that the RBD is not the only region that actively participates in inducing NAbs against SARS-CoV-2 [49], like the NTD. While the function of the NTD as a neutralization target is not yet fully understood, it has been observed that other NTD-specific neutralizing antibodies, in MERS, can inhibit the conformational change from prefusion to postfusion even after receptor binding occurs [9,50].

This study has potential limitations. The number of volunteers at 180 dpv was limited, reducing the strength of the comparison for this time point. An analysis of the cellular immune response can help to correlate the lasting humoral response at 180 dpv with T cell activation [51] and additional isotype assays can elucidate the potential contribution of, in particular, serum IgA [52] in neutralization and lasting humoral response.

In conclusion, the vaccine immunogenicity reported in this study is in agreement with the high efficacy reported for the Sputnik V vaccine [20,23], and this vaccine is able to induce immunity lasting for at least 180 days. The dissection of the Ab reactivity to different regions of S allowed us to identify the relevance of epitopes outside the RBD that are able to induce NAbs. We suggest that the polyclonal and diverse responses of antibodies against S could confer different levels of protection, including neutralizing antibodies and a lasting response.

## Figures and Tables

**Figure 1 antibodies-13-00041-f001:**
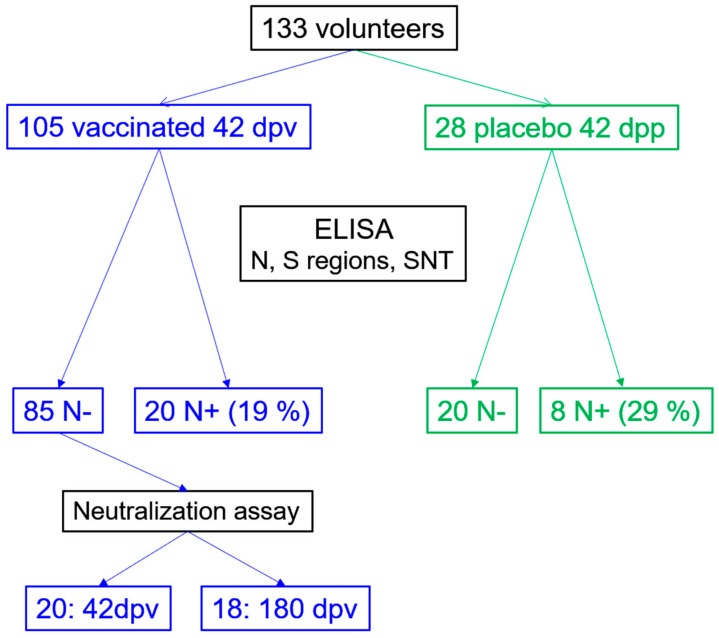
Sputnik V clinical trial group. The stratification of the sample set based on the reactivity results for SARS-CoV-2 N (exposure) is presented. Reactivity was determined in a total of 133 samples (vaccinated and placebo) at 42 dpv. Then, 21 samples of vaccinated and N-negative were selected to compare their reactivity at 42 and 18 to 180 dpv. From these, 20 sera were selected for neutralization assays and comparison between 42 and 180 dpv. SNT: surrogate neutralization.

**Figure 2 antibodies-13-00041-f002:**
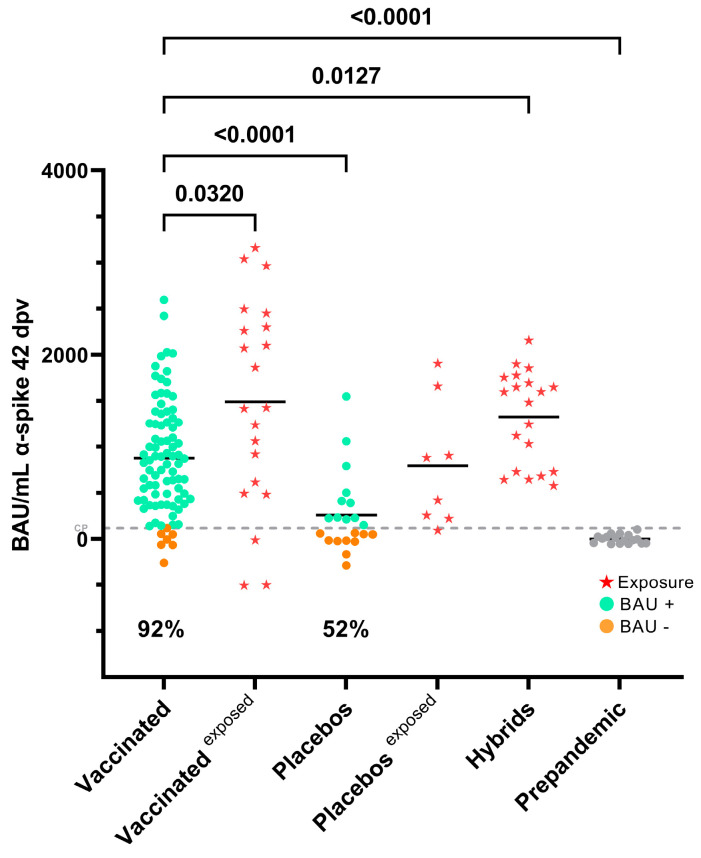
BAU reactivity to S. The different groups are shown: vaccinated, exposed vaccinated, placebos and exposed placebos (classification made according to the reactivity to N), in addition to hybrids and prepandemic. The percentage values represent individuals whose BAU values exceeded the cutoff point (CP). The Kruskal–Wallis test (*p* < 0.0001) with Dunn’s correction post hoc analysis was used for group comparisons. The dashed line indicates the cutoff value (119 BAU/mL).

**Figure 3 antibodies-13-00041-f003:**
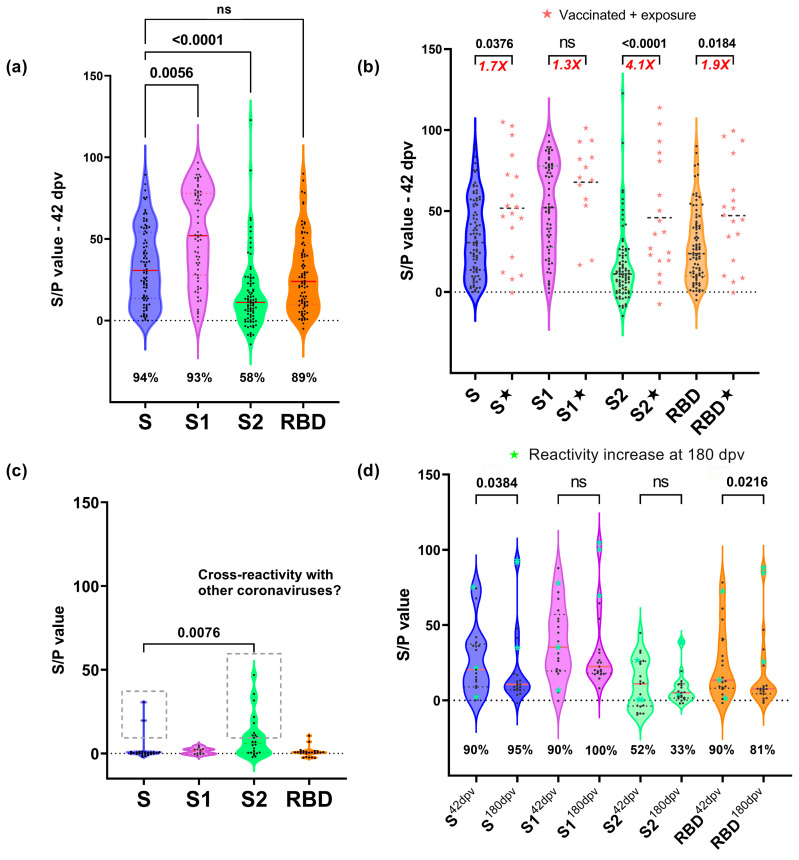
Reactivity towards different regions of S. (**a**) Multiple comparisons of the reactivity of sera from vaccinated individuals at 42 dpv by the Kruskal–Wallis test (*p* < 0.0001) with Dunn’s correction post hoc analysis. The *Y*-axis represents the percent of sample-to-positive ratio value. For S1, only 61 samples are plotted. Percentages of responders are shown. (**b**) Mann–Whitney test of sera from individuals vaccinated with Sputnik-V and exposed. The reactivity toward the different regions of S is shown compared to individuals who were vaccinated and had apparent exposure to the virus during the clinical trial (★); the increased S/P value between the vaccinated and the exposed vaccinated is shown. (**c**) Reactivity of prepandemic sera towards the different regions of S were analyzed by the Kruskal–Wallis test (*p* < 0.0001) with Dunn’s correction post hoc. (**d**) Paired Wilcoxon test reactivity toward the different regions of the spike (ancestral) at 42 and 180 dpv in the cohort that completed the clinical trial without N reactivity (2 samples excluded). ns: not significant.

**Figure 4 antibodies-13-00041-f004:**
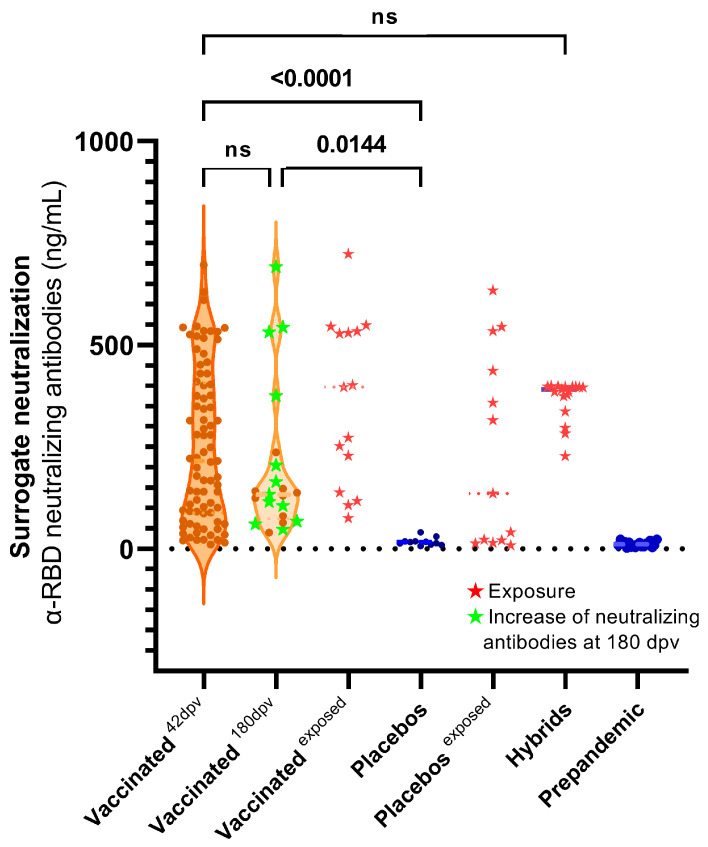
SNT (NAb α-RBD) at 42 and 180 days post-vaccination. NAb titers (ng/mL) at 42 and 180 dpv. Wilcoxon was performed only for paired sera with 42 and 180 dpv samples. ★ Red: two vaccinated individuals with evidence of exposure (excluded from the statistics). ★ Green: sera for which the SNT levels increased at 180 days. The Kruskal–Wallis test (*p* < 0.0001) with Dunn’s correction post hoc was performed for the other comparisons. ns: not significant.

**Figure 5 antibodies-13-00041-f005:**
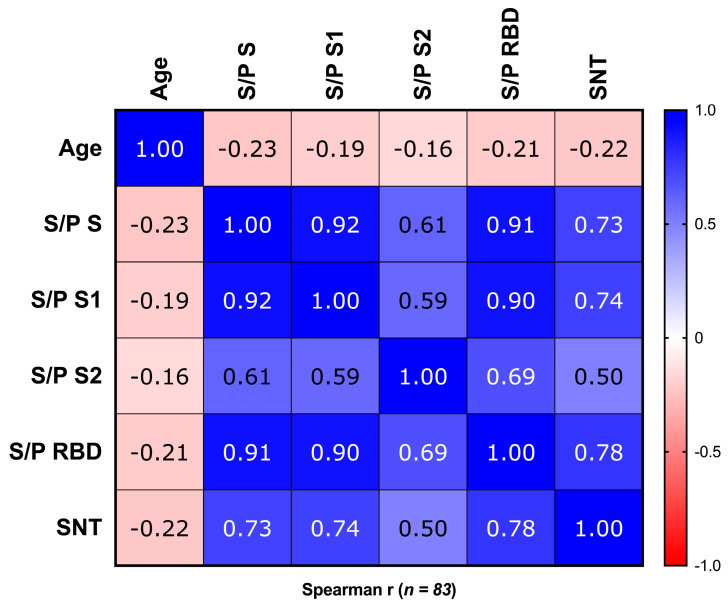
Correlation matrix between reactivity, surrogate neutralization and ages at 42 dpv. Association between variables is represented with a heatmap where blue and red are used to represent positive and negative correlations, respectively. For S1, only 61 samples of S1 were used in the correlations. S/P mean values for S1, S2 and RBD do not show significant correlations with age; the rest of the correlations show *p* values under 0.05.

**Figure 6 antibodies-13-00041-f006:**
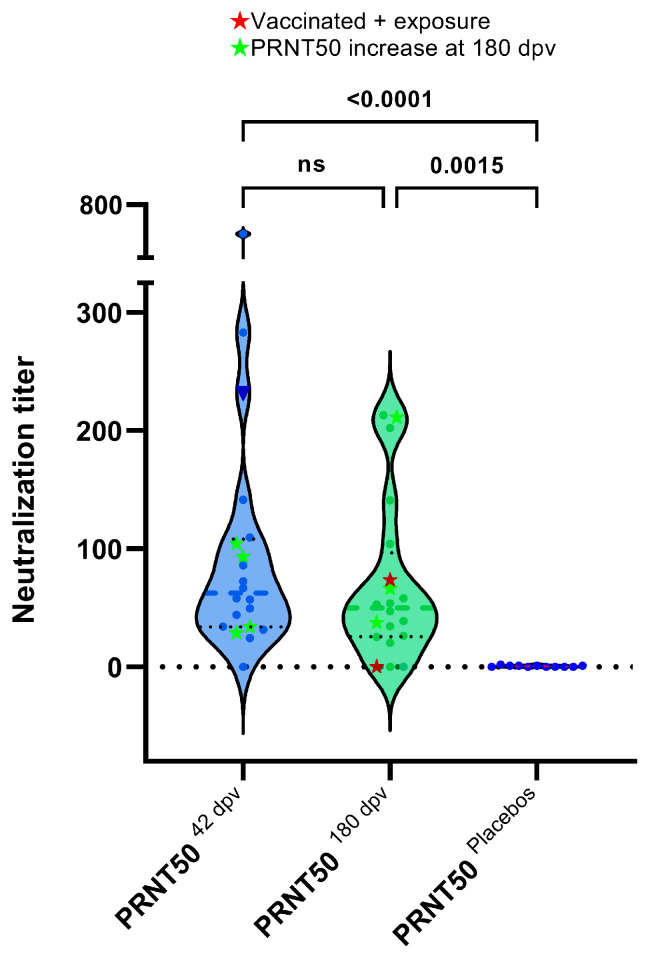
PRNT50 values of the sera from vaccinated individuals at 42 and 180 dpv and from nonexposed placebos at 42 dpv. The *Y*-axis corresponds to the PRNT50 of each serum. ★ Red: two vaccinated individuals with evidence of exposure (excluded from the statistics). ★ Green: sera for which the PRNT50 titer increased at 180 days. Kruskal–Wallis test (*p* < 0.0001). ns: not significant.

**Figure 7 antibodies-13-00041-f007:**
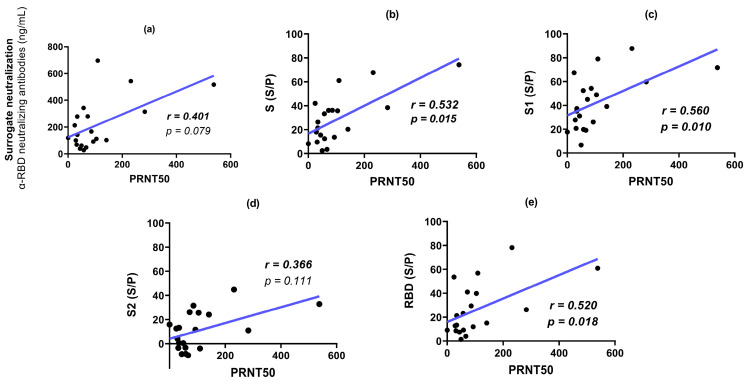
Correlations between PRNT50 titers, SNT (NAb α-RBD) and ELISA reactivity to S regions in individuals vaccinated with Sputnik-V (42 dpv). (**a**): SNT vs. PRNT50. (**b**): Reactivity to S vs. PRNT50. (**c**): Reactivity to S1 vs. PRNT50. (**d**): Reactivity to S2 vs. PRNT50. (**e**): Reactivity to RBD vs. PRNT50. r: Spearman correlation coefficient and *p* value (*p* < 0.05); 95% CI; linear regression model shown as a blue line.

**Figure 8 antibodies-13-00041-f008:**
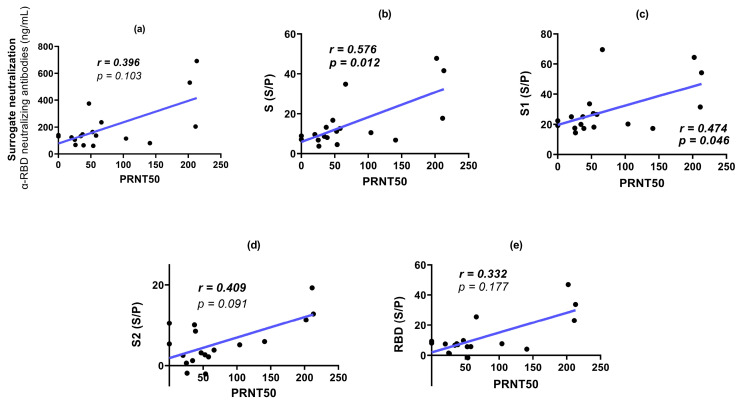
Correlations between PRNT50 titers, SNT (NAb α-RBD) and ELISA reactivity to S regions in individuals vaccinated with Sputnik-V (180 dpv). (**a**): SNT vs. PRNT50. (**b**): Reactivity to S vs. PRNT50. (**c**): Reactivity to S1 vs. PRNT50. (**d**): Reactivity to S2 vs. PRNT50. (**e**): Reactivity to RBD vs. PRNT50. r: Spearman correlation coefficient and *p* value (*p* < 0.05); 95% CI; linear regression model shown as a blue line.

## Data Availability

The complete genome sequence of the SARS-CoV-2 strain used for neutralization assays has been deposited in the GISAID database (accession number EPI_ISL_6980947).

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
