# Peer review of "Humoral Immunity across the SARS-CoV-2 Spike after Sputnik V (Gam-COVID-Vac) Vaccination"

_2073-4468, 2024, doi:10.3390/antib13020041_

Round 1

Reviewer 1 Report

Comments and Suggestions for Authors

Dear authors,

I find your manuscript Humoral immunity across the SARS-CoV-2 Spike after Sputnik  V (Gam‑COVID‑Vac) vaccination

is useful to identify the utility and importance of vaccination in COVID-19.

The introduction and discussion section cover in my opinion sufficient information. Material and methods are well structured and reach all technical requirements. I believe the results are presented properly.

The research design of the study is appropriate and the cited references are adequate and relevant for the research.

Because all of these I believe your manuscript is ready for publication.

Author Response

We thank the reviewer for the comment.

Reviewer 2 Report

Comments and Suggestions for Authors

The authors investigate the antibody response generated after vaccination with two doses of the Sputnik vaccine, in terms of binding to different portions of the Spike protein and of short-term and long-term neutralization. Although there is no significant difference in anti-spike IgG titers between the vaccinated group and the placebo group (both without exposure), the authors show that serum from vaccinated individuals up to 180 days after vaccination has significant neutralizing capacity compared to the placebo group, demonstrating that the vaccine provides a long-lasting protection.

Major concerns:

1. The authors explore the reactivation of sera of individuals in terms of IgG binding (Figures 2 and 3). Several studies have shown the importance of binding and neutralization IgA, even measured in serum, for SARS-CoV-2 infection (DOI: 10.1126/scitranslmed.abd2223. DOI: 10.1084/jem.20220638.). The authors could investigate the IgA response conferred by the Sputnik vaccine.

2. The Sputnik vaccine uses the Spike protein derived from the ancestral variant (B.1) as an immunogenic target. Many articles have shown a reduction in the neutralizing capacity of serum, from vaccinated individuals, against other variants (DOI: 10.1016/j.cell.2021.03.013. DOI: 10.1016/j.ebiom.2022.104158). The authors should investigate the neutralizing capacity of the sera, 180 dpv, against other variants of concern, such as B.1.1.7, B.1.351, P.1 or Omicron.

3. In lines 159-160, the authors write that the PRNT was conducted according to reference number 14. However, the article corresponding to this reference does not perform any neutralization analysis, therefore, the methodology of any PRNT is not explained. Authors must adequately describe the PRNT assay used and check the reference. The authors must also explain some important methodological information, such as: the incubation time of the culture after infection plus antibodies; before fixation; for staining and counting the formed plaques; whether the antibodies were previously incubated with the virus and what is the time of this incubation; how long the inoculum was incubated with the cells before adding the semi-solid medium.

Minor concerns:

1. In Figure 6, it is not clear whether the PRNT50 of the placebos that is compared refers to the time of 30 dpv or 180 dpv.

Comments on the Quality of English Language

I am not an English speaker, but minor editions would improve reading.

Author Response

The authors investigate the antibody response generated after vaccination with two doses of the Sputnik vaccine, in terms of binding to different portions of the Spike protein and of short-term and long-term neutralization. Although there is no significant difference in anti-spike IgG titers between the vaccinated group and the placebo group (both without exposure), the authors show that serum from vaccinated individuals up to 180 days after vaccination has significant neutralizing capacity compared to the placebo group, demonstrating that the vaccine provides a long-lasting protection.

We thank the reviewer for the comment, and apologize of the omission in Figure 2: the difference between vaccinated and Placebos (non-exposed) is indeed significant (p<0.001). This difference is now included in the edited Figure 2 and mentioned in text.

Major concerns:

  1. The authors explore the reactivation of sera of individuals in terms of IgG binding (Figures 2 and 3). Several studies have shown the importance of binding and neutralization IgA, even measured in serum, for SARS-CoV-2 infection (DOI: 10.1126/scitranslmed.abd2223. DOI: 10.1084/jem.20220638.). The authors could investigate the IgA response conferred by the Sputnik vaccine.

We thank the reviewer for his important suggestion. Since we had access to serum and not to saliva, we privileged the analysis of IgG instead of the IgA. In addition, the main objective of this study was to explore the reactivity to the different regions of the S protein, not the relative contribution of the different isotypes in the protective immunity. Finally, IgA antibodies are not expected to be induced preferentially with the Sputnik V vaccination. We include however a comment in this regard, as a limitation of this study (lines 383-384).

  1. The Sputnik vaccine uses the Spike protein derived from the ancestral variant (B.1) as an immunogenic target. Many articles have shown a reduction in the neutralizing capacity of serum, from vaccinated individuals, against other variants (DOI: 10.1016/j.cell.2021.03.013. DOI: 10.1016/j.ebiom.2022.104158). The authors should investigate the neutralizing capacity of the sera, 180 dpv, against other variants of concern, such as B.1.1.7, B.1.351, P.1 or Omicron.

We thank the reviewer for his important comment. We have indeed tested the neutralizing ability of these sera against other variants, which is a matter of a second article on this topic. As stated before, we privileged exploring the reactivity to different regions of the S protein in this study.

  1. In lines 159-160, the authors write that the PRNT was conducted according to reference number 14. However, the article corresponding to this reference does not perform any neutralization analysis, therefore, the methodology of any PRNT is not explained. Authors must adequately describe the PRNT assay used and check the reference. The authors must also explain some important methodological information, such as: the incubation time of the culture after infection plus antibodies; before fixation; for staining and counting the formed plaques; whether the antibodies were previously incubated with the virus and what is the time of this incubation; how long the inoculum was incubated with the cells before adding the semi-solid medium.

We apologize for the mistake. The reference of the PRNT methodology was omitted in the first version and is now included (ref 15). In addition, some additional information about the methodology was included as suggested in lines 168-173.

Minor concerns:

  1. In Figure 6, it is not clear whether the PRNT50 of the placebos that is compared refers to the time of 30 dpv or 180 dpv.

The sera from the placebos were from 42 dpv. More information on the placebos was included in the legend of Figure 6.

Comments on the Quality of English Language

I am not an English speaker, but minor editions would improve reading.

The manuscript was carefully read and edited for English accuracy. All the editions are shown in red in the highlighted version of the manuscript.

Reviewer 3 Report

Comments and Suggestions for Authors

The authors investigated the antibody reactivity against the Spike protein S in sera from vaccinees, as participants of a clinical trial in Venezuela, after vaccination with the two-component vector vaccine Gam-COVID-Vac (Sputnik V). They measured neutralizing antibody reactivity correlating them with the reactivity of antibodies to S up to 180 days after vaccination. (dpv). In particular, relevant epitopes outside of the RBD region of S were identified, in line with the interpretation that Sputnik V can induce immunity for at least 180 days.

Some points need to be addressed:

1. Any reference to publications in peer-reviewed scientific journals and from government authorities providing evidence for a minor/moderate health risk by COVID-19 as well as insufficient effectiveness and adverse effects caused by COVID mRNA vaccines are lacking. This is particularly true for the Pfizer/BioNTech mRNA vaccines which are not directly relevant to the present study, but reference is given in the Introduction (l. 42-45 citing ref. 2). This should be complemented with a brief statement that there are serious indications that public health has been hurt by this vaccination campaign. A suitable reference is "Polykretis et al. (2023) Autoimmune inflammatory reactions triggered by the COVID-19 genetic vaccines in terminally differentiated tissues." and references therein, in particular the released Pfizer documents on phmpt.org (forced by a court decision in November 2021 upon a Freedom of information Act request) confirming a low vaccination effectiveness and the occurrence of severe adverse effects.

This reviewer is not aware of a similar scale of such serious health problems caused by Sputnik V - as reported after vaccination with the COVID-19 mRNA vaccines. Nonetheless, the concept of using a gene therapy (via AAV vectors), which is a powerful tool to treat inborn gene defects, as a vaccine is questionable. The more so as there is evidence for widespread toxic properties of the encoded Spike protein used as antigen, e.g. the occurence of Spike poteins in many damaged organs of perished COVID-19 mRNA vaccinees found in autopsies.

2. l. 297-300: The authors emphasize that only two of the apparently exposed participants from the placebo group reported a symptomatic COVID-19 episode, suggesting an asymptomatic exposure for the other cases.

The correlation of infections with disease symptoms is essential as it remains otherwise unclear whether an infection is causatively related to any „disease“. (This is particularly relevant as a large part of the COVID-19 infections stated in government pieces of information and many studies are assigned as a consequence of RT-PCR test results which is not suitable as a diagnostic tool, e.g. as explicitly stated by its inventor.) In this respect, it should be noted that the symptoms correlating with „usual“ influenza infections and detoxification reactions are essentially identical to those of the COVID-19 infection and it may be difficult or even not possible in many cases to verify the causative effect of a COVID-19 infection to disease symptoms. This may be briefly discussed.

2a. In line with above comments, RT-PCR results are not a suitable diagnostic tool to prove a COVID-19 infection (e.g. l. 128), as stated among others by the inventor of the RT-PCR method.

3. The authors reconcile their data with the important conclusion that a natural infection may provide a stronger immunity, i.e. a stronger reactivity to S2 (l.346-347). This suggestion should be included in the conclusions at the end of the Discussion.

3a. In line with the benefit of naturally induced immunity, the authors cite as a limitation that cell-mediated immune response has not been addressed (l. 373-374), which should be indicated/discussed as a likely contributor of natural immunity against S.

Comments on the Quality of English Language

Some editing is recommended.

Author Response

The authors investigated the antibody reactivity against the Spike protein S in sera from vaccinees, as participants of a clinical trial in Venezuela, after vaccination with the two-component vector vaccine Gam-COVID-Vac (Sputnik V). They measured neutralizing antibody reactivity correlating them with the reactivity of antibodies to S up to 180 days after vaccination. (dpv). In particular, relevant epitopes outside of the RBD region of S were identified, in line with the interpretation that Sputnik V can induce immunity for at least 180 days.

Some points need to be addressed:

  1. Any reference to publications in peer-reviewed scientific journals and from government authorities providing evidence for a minor/moderate health risk by COVID-19 as well as insufficient effectiveness and adverse effects caused by COVID mRNA vaccines are lacking. This is particularly true for the Pfizer/BioNTech mRNA vaccines which are not directly relevant to the present study, but reference is given in the Introduction (l. 42-45 citing ref. 2). This should be complemented with a brief statement that there are serious indications that public health has been hurt by this vaccination campaign. A suitable reference is "Polykretis et al. (2023) Autoimmune inflammatory reactions triggered by the COVID-19 genetic vaccines in terminally differentiated tissues." and references therein, in particular the released Pfizer documents on phmpt.org (forced by a court decision in November 2021 upon a Freedom of information Act request) confirming a low vaccination effectiveness and the occurrence of severe adverse effects.

          Since the aim of this study was to analyze the Ab response to Sputnik V vaccination, we did not address the eventual adverse effects observed with other vaccines.

This reviewer is not aware of a similar scale of such serious health problems caused by Sputnik V - as reported after vaccination with the COVID-19 mRNA vaccines. Nonetheless, the concept of using a gene therapy (via AAV vectors), which is a powerful tool to treat inborn gene defects, as a vaccine is questionable. The more so as there is evidence for widespread toxic properties of the encoded Spike protein used as antigen, e.g. the occurence of Spike poteins in many damaged organs of perished COVID-19 mRNA vaccinees found in autopsies.

We included at the beginning of results, a mention in the absence of severe adverse effects during this clinical trial (lines 191-193).

  1. l. 297-300: The authors emphasize that only two of the apparently exposed participants from the placebo group reported a symptomatic COVID-19 episode, suggesting an asymptomatic exposure for the other cases.

The correlation of infections with disease symptoms is essential as it remains otherwise unclear whether an infection is causatively related to any „disease“. (This is particularly relevant as a large part of the COVID-19 infections stated in government pieces of information and many studies are assigned as a consequence of RT-PCR test results which is not suitable as a diagnostic tool, e.g. as explicitly stated by its inventor.) In this respect, it should be noted that the symptoms correlating with „usual“ influenza infections and detoxification reactions are essentially identical to those of the COVID-19 infection and it may be difficult or even not possible in many cases to verify the causative effect of a COVID-19 infection to disease symptoms. This may be briefly discussed.

Since we do not have any confirmation of exposure for these volunteers, we used term ¨suggest¨ in text.

2a. In line with above comments, RT-PCR results are not a suitable diagnostic tool to prove a COVID-19 infection (e.g. l. 128), as stated among others by the inventor of the RT-PCR method.

          We just suggest a possible exposure based on antibody reactivity. For positive hybrid controls, we include that they were also symptomatic (lines 128-129).

  1. The authors reconcile their data with the important conclusion that a natural infection may provide a stronger immunity, i.e. a stronger reactivity to S2 (l.346-347). This suggestion should be included in the conclusions at the end of the Discussion.

          We have mentioned that hybrid immunity is associated with a stronger immune response. A reference to hybrid immunity was included in lines 130-132.

3a. In line with the benefit of naturally induced immunity, the authors cite as a limitation that cell-mediated immune response has not been addressed (l. 373-374), which should be indicated/discussed as a likely contributor of natural immunity against S.

          We did not address in this study immunity provided by natural infection, except in the context of hybrid immunity.

Comments on the Quality of English Language

Some editing is recommended.

The manuscript was carefully read and edited for English accuracy. All the editions are shown in red in the highlighted version of the manuscript.

Round 2

Reviewer 2 Report

Comments and Suggestions for Authors

The authors addressed all the concerns, and I feel It could be accepted for publication.